# Pancreatic Stone Protein: Review of a New Biomarker in Sepsis

**DOI:** 10.3390/jcm11041085

**Published:** 2022-02-18

**Authors:** Pedro Fidalgo, David Nora, Luis Coelho, Pedro Povoa

**Affiliations:** 1Polyvalent Intensive Care Unit, Hospital São Francisco Xavier, CHLO, 1449-005 Lisbon, Portugal; davidjgarcianora@gmail.com (D.N.); luismiguelcoelho16@gmail.com (L.C.); pedrorpovoa@gmail.com (P.P.); 2Nova Medical School, Clinical Medicine, CHRC, New University of Lisbon, 1169-056 Lisbon, Portugal; 3Research Unit of Clinical Epidemiology, Center for Clinical Epidemiology, Odense University Hospital (OUH), 5000 Odense, Denmark

**Keywords:** pancreatic stone protein, sepsis, infection, biomarker

## Abstract

Sepsis is a life-threatening syndrome characterized by a dysregulated host response to an infection that may evolve rapidly into septic shock and multiple organ failure. Management of sepsis relies on the early recognition and diagnosis of infection and the providing of adequate and prompt antibiotic therapy and organ support. A novel protein biomarker, the pancreatic stone protein (PSP), has recently been studied as a biomarker of sepsis and the available evidence suggests that it has a higher diagnostic performance for the identification of infection than the most used available biomarkers and adds prognostic value. This review summarizes the clinical evidence available for PSP in the diagnosis and prognosis of sepsis.

## 1. Introduction

Sepsis is defined as life-threatening organ dysfunction caused by a dysregulated host response to infection [1]. Although sepsis incidence and mortality seems to be decreasing worldwide, it still represents a total of 19.7% (18.2–21.4) of all global deaths [2], and is the leading cause of in-hospital death and hospital readmission, as well the most expensive hospital condition to treat [3,4]. Despite considerable improvements in the management of sepsis, including early administration of adequate antibiotic therapy and support of organ dysfunction, mortality rates still remain high and early recognition of sepsis is essential and a major determinant of the disease’s outcome [5,6,7].

The lack of a gold standard test to diagnose infection as well as the overly sensitive and nonspecific features of signs and symptoms of sepsis led medical societies to endorse the use of biomarkers (“inflammatory variables”) as surrogate markers of infections to help clinicians in its diagnosis [8]. However, in clinical practice, the diagnosis of infection still relies on the intersection of three vectors: systemic manifestations, organ dysfunction and microbiological documentation [9], and no single biomarker or diagnostic test, per se, has been validated to diagnose infection.

The weaknesses of the current framework used to diagnose infection and sepsis are illustrated by several examples: the disagreement between sepsis diagnosis at the intensive care unit or emergency department admission and posthoc assessment [10,11], leading to erroneous treatment of >40% of patients as septic with an unlikely infection [12], or the inadequate antibiotic prescription for patients admitted with viral diseases (e.g., influenza) [13]. There is a growing need for fast and adequate infectious disease diagnostic procedures [14], although special attention should be focused on the features of an ideal diagnostic test—ASSURED—affordable, sensitive, specific, user-friendly, rapid, equipment-free, and delivered to those in need [15,16].

The majority of biomarkers used in sepsis assess prognosis [17]. A good predictive biomarker of infection, however, should be absent if the patient is not infected, should appear concomitantly with and ideally preceding the clinical manifestations of infection, and disappear with successful therapy or remain elevated if infection is refractory to treatment [18,19]. In the context of septic shock, the association between delay in antibiotic administration and death seems stronger than in septic patients without shock [20] supporting the recommendation to administer antimicrobials within one hour in all patients with septic shock. Therefore, point-of-care testing is appealing, as it might provide clinicians with a rapid and readily available diagnosis.

## 2. Materials and Methods

The selection of studies to describe pancreatic stone protein function, diagnostic and prognostic ability was conducted according to the Preferred Reporting Items for Systematic Reviews and Meta-analyses (PRISMA) guidance [21]—Figure 1.

Relevant studies up to October 2021 were searched in Pubmed and Cochrane Library databases with the terms “pancreatic stone protein”, “sepsis biomarker(s)” and their combination. Moreover, references of the retrieved manuscripts were also manually cross-searched for further relevant publications. The inclusion criteria were as follows: (1) studies including adult (>18 years old) patients, and (2) studies published with full-text. The exclusion criteria were as follows: (1) studies using data retrieved post mortem. (2) clinical trial protocol.

## 3. Pancreatic Stone Protein (PSP): Structure, Function and Kinetics

Lithostathine and regenerating protein 1 (Reg I) were described by different groups working on pancreatitis and diabetes during the decade of the 1980s [22]. Later on, both proteins were found to be structurally identical, synthesized in the pancreatic acinar cells as a single polypeptide and secreted into the duodenum along the same secretory pathway as the exocrine enzymes. Therefore, they were renamed as pancreatic stone protein, since its first attributed function was (inaccurately) thought to be the inhibition of calcium carbonate crystals precipitation in the pancreatic juice [23,24]. Later on, the discovery of PSP in other organs besides the pancreas (e.g., brain) [25,26] and the discovery of its functional antibacterial activity [27] led investigators to explore whether it could be involved in other processes besides solubilization of the pancreatic content. 

Nowadays it is established that PSP is a 14 kDa insoluble polypeptide encoded by a single transcript of the *reg* gene, resulting in a 144-amino acid length glycoprotein, structurally similar to C-type lectin-like proteins, [28] which are calcium-dependent glycan-binding proteins involved in the process of cell to cell and host-cell interaction, including adhesion and signaling receptors in homeostasis and innate immunity as well leukocyte and platelet trafficking in inflammatory responses [29]. PSP levels were shown to be slightly higher in patients with Type-2 diabetes mellitus compared with healthy individuals [30], being significantly higher in the subset of patients with diabetic kidney disease [31], probably due to a filtration effect suggesting renal dysfunction [32].

To determine its biological and functional role, a pivotal observation was accidently made in rat experiments by the group of Rolf Graf in which PSP was found to be an indicator of systemic stress [33]. This observation was then clinically confirmed by the demonstration in humans that the pancreas senses remote organ damage and systemic stress and responds by secreting PSP in the absence of pancreatic tissue damage [34]. As an acute-phase protein, PSP might be involved in promoting cell proliferation during regenerative processes [35], through regulation by IL-6 and other cytokines that are released after tissue injury [36,37], rendering to the pancreas what Reding et al. [38] call “the role of an acute phase organ”.

The role of PSP in the immune and inflammatory response to infection prompted its identification as a potential biomarker of infection and sepsis.

The evaluation of PSP has evolved from conventional laboratory methods, such as the isoform-specific enzyme-linked immunosorbent assay (ELISA) using the sandwich technique [34], to point of care methods at the patient’s bedside [39]. The latter underwent analytical validation and maintained a reliable performance [39,40] with faster results, which is particularly appealing in the diagnosis of septic patients, for whom speed of intervention is crucial for the prognosis [1].

## 4. PSP Performance for the Diagnosis of Infection and Sepsis

The performance of PSP as a biomarker of infection and sepsis has been evaluated in several patient populations and clinical settings [34,41,42,43,44,45,46,47]—Table 1.

Although the majority of available studies used the 2001 definitions of sepsis and infection [8], overall the performance of PSP discriminating infection/sepsis vs. no infection/sepsis is at least comparable to other canonical biomarkers of infection and might even be better in some particular situations. Gukasjan et al. [48] found significantly higher PSP levels at ICU admission [15.2 (11.2–23.2) ng/mL vs. 125.0 (25.0–419.0) ng/mL] in patients with secondary peritonitis, compared to a control group of 43 patients admitted for elective surgery. After cardiac surgery, PSP performed better than CRP and white blood cell count for the diagnosis of infection [43].

Klein et al. [49] showed that in a cohort of burn patients admitted to the ICU without sepsis, the serum levels of PSP remained unchanged over time not only after the initial burn injury but also after secondary debridement procedures in contrast to CRP and PCT both of which significantly increased after inflammatory and/or surgical insults, suggesting that PSP might be a more robust biomarker of sepsis in this particular setting. In another cohort of burn patients admitted to the ICU, PSP demonstrated a 3.3–5.5-fold increase for up to 72 h before the diagnosis of sepsis [50] and among those with inhalation injury and ARDS, PSP was the strongest marker to identify sepsis when compared to CRP and PCT both by its higher values and steeper increase over time [51].

Scherr et al. [52] showed that among patients admitted to the emergency department with exacerbation of chronic obstructive pulmonary disease, PSP levels were significantly higher among those with positive sputum cultures compared to those with negative sputum cultures at exacerbation and those with stable disease.

Finally, Prazac et al. [53] have recently conducted an individual patient level meta-analysis and found PSP to perform better than CRP or PCT for the diagnosis of community-acquired infections in the emergency department and surgical infections after cardiac surgery.

PSP demonstrates a significant interaction between time and presence of sepsis [46,47], suggesting that besides a fixed cut-off value (as in standard ROC curve analysis) the time-related kinetics of PSP has a crucial role in the identification of sepsis when considering the time-dependency of the infectious/septic event. CRP had also shown usefulness in the timely stratification of the risk of infection in critically ill patients (patients presenting maximum daily CRP variation >4.1 mg/dL plus a CRP level >8.7 mg/dL had an 88% risk of ICU-acquired infection [54]), in prediction of VAP in the first six days of mechanical ventilation (rate of CRP change per day, highest level and maximum amplitude of variation were all significantly associated with VAP development [55]), and in anticipation of community-acquired bloodstream infection (CRP concentrations began to increase 3.1 days before diagnosis [56]). Such an approach of time-profiling a biomarker may be more helpful, informative and accurate [47]. According to Pugin et al. [47] PSP outperformed the other classic biomarkers by its relative increase even five days before clinical diagnosis of sepsis compared to three days for PCT and two days for CRP.

## 5. PSP Performance for the Prognosis of Septic Patients

In addition to its usefulness in the diagnosis of infection, PSP has shown a good performance in the prognosis of septic patients. Table 2 describes the main characteristics of studies on the prognostic value of PSP.

Boeck et al. [57] retrospectively evaluated PSP in a cohort of 101 patients with VAP, and found significantly higher values in non-survivors both on the day of diagnosis and on day 7, with different predictive mortality thresholds at each time point.

Que et al. [58] prospectively analyzed the serum value of a set of biomarkers (PSP, CRP, PCT, IL-6, IL-8, IL-10, TNF-α and IL-1 ß) in 107 patients with severe sepsis and septic shock (according to the Sepsis 2 criteria) in the first 24 h after ICU admission. PSP was the only analysed biomarker significantly increased in non-survivors.

Guadiana-Romualdo et al. [59] evaluated the prognostic ability of PSP in septic patients. PSP was measured in the first 6 h after diagnosis (baseline) and on the second day of admission to the ICU in 122 patients. It was found not only that PSP was significantly higher in non-survivors at both measurement times but also that there was a decreasing trend of PCT between measurements in the group of survivors.

In patients admitted to the ICU in the immediate postoperative period of abdominal surgery for secondary peritonitis (*n* = 91), PSP assessed at admission was the only biomarker (compared to CRP, PCT, IL-6 and WBC) with the ability to discriminate between clinical severity and predict mortality [48].

Que et al. [60] analysed two cohorts of ICU septic patients (total *n* = 249) to assess the prognostic value of PSP and to validate a mortality predictive model using severity scores and biomarkers. Higher PSP values were associated with clinical severity (significantly in both cohorts) and non-survivors (reaching statistical significance in only one cohort). Models with the addition of biomarkers (PSP, CRP and PCT) with severity indexes showed a better predictive capacity for in-hospital mortality than each parameter individually.

More recently, in a population of SARS-CoV2-infected patients admitted to the emergency department, PSP was higher in non-survivors but was not accurate to discriminate patients with organ dysfunctions that required admission to the ICU [61].

## 6. Clinical Application

### 6.1. Emergency Department

In the reality of the emergency department, PSP can be useful for the early diagnosis of infection and for the triage of patients based on the risk of mortality. The diagnostic ability of PSP may be relevant not only through its sensitivity for timely diagnosis, but also through its negative predictive value, which can lead to a reduction in inappropriate antibiotic prescriptions, which in compliance with antibiotic stewardship strategies. The importance of a triage based on analytical clinical data (such as a biomarker) has been well demonstrated in recent years, in which due to the context of the SARS-CoV2 pandemic, it has become more fundamental than ever to manage resources and ensure that existing resources are best adapted to patients’ needs.

### 6.2. Intensive Care Unit

In intensive care units, single assessments would predominantly have a prognostic value at admission. This data would make it possible to optimize the allocation of resources and can serve as a quality assessment and benchmarking tool (as is already the case with some severity indices, such as APACHE IV and SAPS 3 scores [62]). The possibility of performing serial assessments of PSP in ICUs would allow for a sentinel effect of infection in patients hospitalized for non-infectious causes and/or monitoring infection response to antibiotic therapy.

## 7. Data Analysis, Limitations and Questions to Be Clarified

It is not yet clear whether PSP assessment is more useful when performed at specific times (at admission or when there is any clinical suspicion of infection), or serially. It is interesting that, in published trials, quite different thresholds for both diagnosis and prognosis were identified. Without well-defined thresholds, the interpretation of the PSP value in single measurements becomes more complex when compared to the analysis of the PSP trend during hospitalization. The association of PSP value with other biomarkers or with clinical severity indices (eventually included in decision and clinical intervention algorithms) represents a new and interesting strategy to overcome the limited prognostic performance of single parameters, but it is also very conditioned when a well-defined risk threshold does not exist.

The relationship of PSP with the presence and severity of organ dysfunctions is another factor to be clarified and which may determine the potential of PSP to stratify patients early according to disease severity, alone or in combination with other scores or indicators. Furthermore, it is equally important to know the changes in PSP kinetics in patients with invasive organ support (such as dialysis and other extracorporeal techniques), and to understand to what extent such changes modify its diagnostic and prognostic usefulness.

It is expected that in the short term the feasibility, clinical utility, and economic benefit of real-time measurements of PSP using point-of-care technology (versus conventional laboratory measurement) will be confirmed. This will facilitate further studies leading to a better and more complete understanding of PSP kinetics in different patients and settings (for example, gram-positive and gram-negative bacteria, viral, fungal and/or parasitic infections). Hopefully, such data will translate into an optimized approach to septic patients (by selecting the intervention that most suits each patient’s condition) and thus contribute to a better outcome.

The applicability and value of PSP in non-infectious circumstances remains to be investigated. As a positive acute-phase protein and systemic stress marker, it will be important to understand its behavior and potential usefulness (diagnostic, prognostic, severity marker) in non-infectious inflammatory circumstances such as trauma and pancreatitis or more generally in other pancreatic diseases.

## 8. Conclusions

PSP accuracy for the diagnosis of infection and sepsis among a wide spectrum of clinical settings seems to be, at least, comparable to the other classical biomarkers currently used in clinical practice. Furthermore, it seems to outperform those biomarkers in the prediction of sepsis, accounting for its earlier relative increase before clinical diagnosis, and it adds prognostic value.

## Figures and Tables

**Figure 1 jcm-11-01085-f001:**
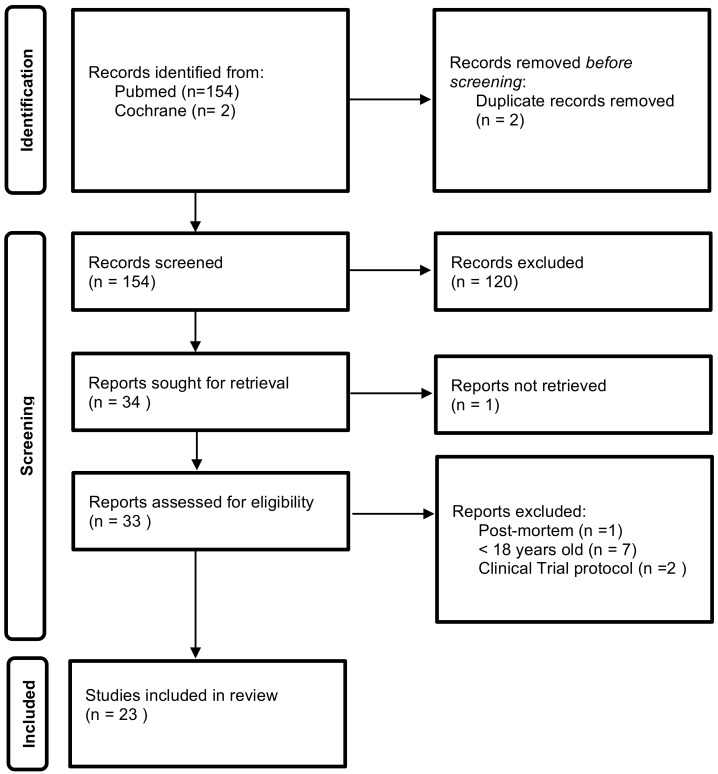
Identification of studies flow diagram.

**Table 1 jcm-11-01085-t001:** Characteristics of studies evaluating PSP diagnostic performance for infection and/or sepsis. ED—Emergency department; ICU- Intensive Care Unit; SIRS—Systemic Inflammatory Response Syndrome; PSP—pancreatic stone protein; sCD25—soluble CD25; PCT—procalcitonin; HBP—heparin binding protein; CRP—C-reactive protein; IL6—interleukin-6; WBC—White blood count; AUC ROC—areas under receiver operating characteristic curves; IQR—interquartile range.

Study Main Features	Population and Objectives	Main Results	Comments
Guadiana-Romualdo et al. [41] Prospective, single-center observational Serum PSP determined on admission to the ED Infection—clinically relevant positive bacterial microbiological cultures collected within 48 h of enrolment or patients with strong evidence (radiographic evidence or physical examination) for infection in the absence of positive cultures. PSP measured by isoform-specific ELISA using the sandwich technique	152 unselected adults (>14 years) patients admitted to the ED with suspicion of infection Primary Objective: comparison of the performance between PSP, sCD25 and PCT for the diagnosis of infection and sepsis	No differences between PCT, sCD25 and PSP discriminative ability for infection (vs. non-infection) or sepsis (vs. non/sepsis) PSP AUC ROC (95% CI) 0.84 (0.77–0.90) for a cut-off of 41.5 ng/mL—infection. PSP AUC ROC (95% CI) 0.87 (0.81–0.94) for a cut-off of 96.6 ng/mL—sepsis.	84.9% of patients with infection. Most common sources were urinary (41.1%) and respiratory tract (31.8%). Infection was microbiologically proven in 53.5%
Keel M et al. [34] Retrospective, single-center, observational. Serum PSP levels were determined at days 0, 1, 3, 5, 7, 10, 14 and 21. Patients were categorized post hoc into three groups: (a) no infection (b) infection without sepsis, (c) sepsis. Sepsis—all four criteria of SIRS were met for three consecutive days in the presence of a septic focus with positive bacterial tissue culture or a positive blood culture. Local infection -If less than four SIRS criteria were observed over three days in the presence of a positive focus PSP measured by isoform-specific ELISA using the sandwich technique	83 trauma adult (>16 years) patients admitted to ICU Primary objective: comparison of PSP levels between groups: infection without sepsis or sepsis vs. noninfected patients and local infection vs. sepsis	PSP increased from 10.5 in all groups to 22.8 ng/mL in patients without infection vs. 111.4 ng/mL in patients with infection without sepsis and 146.4 ng/mL in septic patients (days 5–10), *p* < 0.05 for comparisons	Grading increase in PSP levels for non-infected, infection without sepsis and septic patients at day five.
Llewelyn et al. [42] Prospective, multicenter, observational. Serum PSP levels were determined during the first six hours of admission. Sepsis—SIRS plus either proven infection (on the basis of microbiological sampling or radiology) or probable infection (presentation, WBC, CRP, radiology) PSP measured by isoform-specific ELISA using the sandwich technique	219 unselected adult patients admitted to ICU or high-dependency unit. Primary objective: Comparison of the performance between PSP and HBP for the diagnosis of sepsis	No difference between the discriminative ability of biomarkers. PSP AUC ROC (95% CI) 0.93 (0.89 to 0.97) for a cut-off 30 ng/mL	43.9% of patients were classified as septic. Most common sources of infection were respiratory tract (38%) or abdomen (44%). Infection was microbiologically proven in 38% patients
Klein et al. [43] Prospective, single-center, observational. Serum PSP levels were determined pre-operatively and 24, 48 and 72 h post surgery Infection defined according to 2001 SCCM/ESICM/ACCP/ATS/SIS International Sepsis Definitions Conference [8] PSP measured by isoform-specific ELISA using the sandwich technique	120 adult (>18 years) patients admitted to the ICU after elective cardiac surgery Primary objective: Comparison of the performance between PSP, CRP and WBC for the diagnosis of infection	Significantly higher performance of PSP compared to other biomarkers (CRP and WBC) that failed to differentiate infection from postoperative inflammatory response. PSP AUC ROC (95% CI) 0.77 (0.62–0.89) for a cut-off of 41.5 ng/mL	Infection among 15% of patients. Most common source of infection was pneumonia (44.4%)
Parlato et al. [44] Prospective, multicenter, observational. Serum PSP were determined at inclusion. Sepsis was defined according to 2001 SCCM/ESICM/ACCP/ATS/SIS International Sepsis Definitions Conference [8] PSP measured by ELISA	279 adult patients admitted to the ICU with hypothermia (below 36.0 °C) or hyperthermia (over 38.0 °C) and at least another criterion of SIRS were eligible as soon as the physician considered antibiotic therapy Primary objective: assess the accuracy of 53 circulating biomarkers to discriminate between sepsis and non-septic SIRS	Median (IQR) PSP (ng/mL) levels were significantly higher is septics vs. non-septic SIRS: 123 (65–269) vs. 73 (42–214), *p* = 0.02 PSP AUC ROC (95% CI) 0.63 (0.54–0.71) lower than CRP	Two-thirds of patients diagnosed as having sepsis blindly to the results of biomarkers. Most common source of infection was the lung (69.8%). 25% of septic patients had positive blood cultures No combination of biomarkers improved the diagnostic accuracy of CRP.
Garcia de Guadiana-Romualdo [45] Prospective, single-center, observational. Serum PSP were determined at admission. Infection was defined as a cluster of clinical signs or symptoms and radiological findings of infection without microbiological proof or Microbiologically documented infection, which includes bacteremia, and microbiologically documented local infection without positive blood culture. PSP measured by ELISA	114 episodes among 105 adult (>18 years) patients admitted to the ED with chemotherapy associated febrile neutropenia. Primary objective: Comparison of the performance between PSP, sCD25 and PCT for the diagnosis of infection	Lower discriminative ability of PSP compared to PCT to the diagnosis of infection. PSP AUC ROC (95% CI) 0.75 (0.66–0.84) for a optimal cut-off of 29.0 ng/mL	51.8% of episodes were of infectious origin.
Klein et al. [46] Prospective, single-center, observational. Serum PSP levels were determined daily from admission to day 10. Sepsis defined according to Third International Consensus Definition for Sepsis and Septic Shock (Sepsis-3) PSP measured by isoform-specific ELISA using the sandwich technique	90 adult patients with burns >15% total body surface area admitted to the ICU. Primary objective: Comparison of the performance between PSP, CRP, PCT and WBC for the diagnosis of sepsis during the first 10 days	PSP and PCT outperformed CRP and WBC. Day 7 post-op PSP AUC ROC (95% CI) 0.89 (0.81–0.96) for a cut-off of 60.12 ng/mL	Sepsis among 51% of patients. Most common source of infection was pneumonia (58%)
Pugin et al. [47] Prospective, multicenter, observational Serum PSP levels were determined daily from admission until death or discharge from the ICU or for 30 days Sepsis defined according to Third International Consensus Definition for Sepsis and Septic Shock (Sepsis-3) PSP measured through nanofluidic point-of-care immunoassay; abioSCOPE^®^.	243 adult patients admitted to ICU at risk for nosocomial infection (expected to stay ≥7 days and/or to be mechanically ventilated ≥5 days). Primary objective: Comparison of the performance between PSP, CRP and PCT for the diagnosis of nosocomial sepsis	Similar performance between biomarkers. PSP AUC ROC (95% CI) 0.75 (0.67–0.82) for a cut-off of 290.5 ng/mL	21.8% of patients developed sepsis, the majority originated from the respiratory tract.

**Table 2 jcm-11-01085-t002:** Characteristics of studies evaluating PSP prognostic value in patients with infection and/or sepsis. VAP—ventilator-associated pneumonia; PSP—pancreatic stone protein; SOFA—Sequential organ failure assessment; ICU- Intensive Care Unit; PCT—procalcitonin; CRP—C-reactive protein; IL6—interleukin-6; IL-8—interleuki-8; TNF-α—tumor necrosis factor alpha; IL-1ß—interleukin-1beta WBC—White blood count; AUC ROC—areas under receiver operating characteristic curves; IQR—interquartile range; OR—odds ratio; CI—confidence interval; ELISA—Enzyme-Linked Immunosorbent Assay; APACHEII—Acute Physiology and Chronic Health Evaluation II; SAPSII—Simplified Acute Physiology Score; SE—sensitivity; SP—specificity.

Study Main Features	Population and Endpoints	Main Results	Comments
Boeck et al. [57] Multi-center, retrospective, observational. Serum PSP levels were determined on VAP diagnosis (baseline) and on day seven PSP measured by isoform-specific ELISA using the sandwich technique.	101 adult ICU patients with VAP. Primary endpoint: 28-day mortality.	PSP was significantly higher in nonsurvivors vs. survivors (117 ng/mL vs. 36.3 ng/mL, *p* = 0.011). Baseline PSP and on day 7 were significant predictors of survival (baseline, OR 1.60, 95% CI, 1.07–2.38, *p* = 0.022; day seven, OR 2.36, 95% CI, 1.27–4.39, *p* = 0.007). PSP AUC ROC for mortality/survival on VAP diagnosis and on day seven was 0.69 and 0.76 (95% CI, 0.57–0.80 and 0.62–0.91), respectively.	PSP was associated with severity and organ dysfunction (SOFA score) from VAP diagnosis up to day 7. PSP cut-off of 24 ng/mL at baseline had the highest accuracy to identify survivors. PSP threshold of 177 ng/mL at day seven to determine patients with a poor chance of survival.
Que et al. [58] Single-center, prospective, observational. Blood samples collected at ICU admission for PSP, PCT, CRP, IL-6, IL-8, TNF-α and IL-1ß measurements. PSP measured by isoform-specific ELISA using the sandwich technique	107 septic adult ICU patients Primary endpoint: in-hospital mortality.	PSP was significantly higher in septic shock vs. severe sepsis (343.5 ng/mL vs. 73.5 ng/mL, *p* < 0.001) as well as PCT, IL-6 and IL-8. PSP was the only biomarker with significant differences between nonsurvivors vs. survivors (397 ng/mL vs. 216.1 ng/mL, *p* = 0.02). PCT was the best predictor of mortality between all biomarkers measured (AUC ROC 0.65).	In patients with septic shock, PSP was the only biomarker associated with in-hospital mortality (*p* = 0.049).
Guadiana-Romualdo et al. [59] Single-center, prospective, observational. Blood samples collected at baseline (within 6 h of sepsis diagnosis) and on day 2 for PSP, PCT, CRP and lactate measurements; SOFA score computed daily. PSP measured by isoform-specific ELISA using the sandwich technique	122 septic adult ICU patients. Primary endpoint: 28-day mortality.	Baseline PSP and lactate were significantly higher in nonsurvivors vs. survivors (*p* < 0.001). On day 2 PSP was significantly higher in nonsurvivors vs. survivors (*p* < 0.001). Decreasing trends in PSP and PCT from baseline to day two were significantly higher in nonsurvivors vs. survivors (*p* < 0.001).	Baseline PSP plus lactate: AUC-ROC 0.796. Baseline SOFA: AUC-ROC 0.826. On day 2 PSP: AUC-ROC 0.844. On day 2 SOFA: AUC-ROC 0.923.
Gukasjan et al. [48] Single-center, prospective, observational. Blood samples collected within 3 h after ICU admission for PSP, PCT, CRP, IL-6 and WBC measurements; SOFA score computed daily. PSP measured by isoform-specific ELISA using the sandwich technique	91 adult ICU patients with secondary peritonitis. ICU admission after first abdominal surgery. Primary endpoint: ICU mortality. Secondary endpoint: 90-day mortality,	PSP was significantly higher in more severe situations [no organ dysfunction 24.4 ng/mL, one to three organ dysfunctions 185.9 ng/mL (*p* < 0.001) and more than 3 organ dysfunctions 721.4 ng/mL (*p* = 0.047)] PSP was significantly higher in nonsurvivors vs. survivors (499.4 ng/mL vs. 75 ng/mL, *p* = 0.003).	PSP cut-off for mortality 130 ng/mL (*p* < 0.001, OR 6.192). PSP: AUC-ROC 0.775. 90-day survival: 96% when PSP < 130 ng/mL and 74% when PSP ≥ 130 ng/mL (*p* = 0.015, RR 6.48)
Que et al. [60] Two centers, prospective, observational. Biomarkers measured and severity scores computed either 24 h after ICU admission or 24 h after the diagnosis of sepsis (for ICU patients with other admission diagnosis). PSP measured by isoform-specific ELISA using the sandwich technique	Two cohorts with a total of 249 adult ICU septic patients (158 + 91). Primary endpoint: in-hospital mortality.	PSP was significantly higher in septic shock vs. severe sepsis (323 ng/mL vs. 78.8 ng/mL, *p* < 0.001, *n* = 158 and 184 ng/mL vs. 58.9 ng/mL, *p* = 0.005, *n* = 91) PSP was significantly higher in nonsurvivors vs. survivors (in the larger cohort only) (346.7 ng/mL vs. 209.8 ng/mL, *p* = 0.002)	PSP and severity scores (individually) had moderate accuracy for the prediction of death in both cohorts (PSP AUC ROC 0.665). The best models for in-hospital mortality included PSP plus PCT with either APACHEII (AUC ROC 0.721) or SAPSII (AUC ROC 0.710). PSP AUC ROC 0.665
Van Singer et al. [61] Single-center, prospective, observational. Blood samples collected on admission for PSP and CRP measurements. Bedside clinical severity scores (pSOFA and CRB-65) assessed. PSP measured through nanofluidic point-of-care immunoassay; abioSCOPE^®^.	173 PCR-confirmed SARS-CoV2 infected patients admitted in the emergency department Primary endpoint: 7-day mortality Secondary endpoint: ICU admission	PSP was significantly higher in nonsurvivors vs. survivors (141 ng/mL vs. 70 ng/mL, *p* < 0.001) as well as CRP, qSOFA and CRB-65. PSP performed worser than CRP to predict ICU admission (AUROC 0.51 vs. 0.74, *p* < 0.001)	The combination of clinical scores with biomarkers performed better than each parameter individually. Combination of PSP and CRP did not perform better than biomarkers or clinical scores alone. The best combinations were CRB-65 with CRP (AUROC 0.96) and CRB-65 with PSP (AUROC 0.95).

## Data Availability

Not applicable.

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
