# Peer review of "Pancreatic Stone Protein: Review of a New Biomarker in Sepsis"

_jcm, 2022, doi:10.3390/jcm11041085_

Round 1
Reviewer 1 Report
Unfortunately in 2019 a review on the same exact subject and with the same structure was published (https://www.futuremedicine.com/doi/10.2217/bmm-2018-0194).
I could not find any novel information in this review, since nearly all references (except from one systematic review) are up to 2019.
The only addition if ref 56, the role of PSP in COVID-19.
If you could add more information, in order to differentiate your review from the one already published and make it more novel.
Author Response
1.“Unfortunately in 2019 a review on the same exact subject and with the same structure was published. If you could add more information, in order to differentiate your review from the one already published and make it more novel"
We recognize that the review published in 2019 share common ground with ours, however we have foccused on the clinical applicability of PSP among critically ill patients and excluded papers including children ( < 18 years old). We have added 9 new references in this version and added a methods, clinical application in critically ill patients (emergency department and ICU) and limitations section.
We believe that this uptodated version adds to the body of knowledge on the field of PSP.
2."The only addition if ref 56, the role of PSP in COVID-19.I could not find any novel information in this review, since nearly all references (except from one systematic review) are up to 2019. If you could add more information, in order to differentiate your review from the one already published and make it more novel."
We acknowledge that since 2019 only a few studies have been reported with novel information regarding PSP clinical applicability in the context of infection and sepsis. However, our manuscript included several new references (aside from the systematic review [ref 45] and COVID [ref 61] you mentioned) compared to the previously published review you cited, including studies in different populations and settings.
ICU
[47] Pugin J, Daix T, Pagani JL, Morri D, Giacomucci A, Dequin PF, et al. Serial measurement of pancreatic stone protein for the early detection of sepsis in intensive care unit patients: a prospective multicentric study. Crit Care 2021;25:1–9. doi:10.1186/s13054-021-03576-8.
[55] Parlato M, Philippart F, Rouquette A, Moucadel V, Puchois V, Blein S, Bedos JP, Diehl JL, Hamzaoui O, Annane D, Journois D, Ben Boutieb M, Estève L, Fitting C, Treluyer JM, Pachot A, Adib-Conquy M, Cavaillon JM, Misset B; Captain Study Group. Circulating biomarkers may be unable to detect infection at the early phase of sepsis in ICU patients: the CAPTAIN prospective multicenter cohort study. Intensive Care Med. 2018 Jul;44(7):1061-1070. doi: 10.1007/s00134-018-5228-3. Epub 2018 Jun 30. PMID: 29959455.
Febrile neutropenia
[56] García de Guadiana-Romualdo L, Jiménez-Santos E, Cerezuela-Fuentes P, Español-Morales I, Berger M, Esteban-Torrella P, Hernando-Holgado A, Albaladejo-Otón MD. Analyzing the capability of PSP, PCT and sCD25 to support the diagnosis of infection in cancer patients with febrile neutropenia. Clin Chem Lab Med. 2019 Mar 26;57(4):540-548. doi: 10.1515/cclm-2018-0154. PMID: 30240355.
Burn patients
[46] Klein HJ, Niggemann P, Buehler PK, Lehner F, Schweizer R, Rittirsch D, et al. Pancreatic Stone Protein Predicts Sepsis in Severely Burned Patients Irrespective of Trauma Severity. Ann Surg 2020;Publish Ah.doi:10.1097/sla.0000000000003784.
[48] Klein HJ, Buehler PK, Niggemann P, Rittirsch D, Schweizer R, Waldner M, Giovanoli P, Cinelli P, Reding T, Graf R, Plock JA. Expression of Pancreatic Stone Protein is Unaffected by Trauma and Subsequent Surgery in Burn Patients. World J Surg. 2020 Sep;44(9):3000-3009. doi: 10.1007/s00268-020-05589-w. PMID: 32451625.
[49] Niggemann P, Rittirsch D, Buehler PK, Schweizer R, Giovanoli P, Reding T, Graf R, Plock JA, Klein HJ. Incidence and Time Point of Sepsis Detection as Related to Different Sepsis Definitions in Severely Burned Patients and Their Accompanying Time Course of Pro-Inflammatory Biomarkers. J Pers Med. 2021 Jul 23;11(8):701. doi: 10.3390/jpm11080701. PMID: 34442346; PMCID: PMC8401386.
[50] Klein HJ, Rittirsch D, Buehler PK, Schweizer R, Giovanoli P, Cinelli P, Plock JA, Reding T, Graf R. Response of routine inflammatory biomarkers and novel Pancreatic Stone Protein to inhalation injury and its interference with sepsis detection in severely burned patients. Burns. 2021 Mar;47(2):338-348. doi: 10.1016/j.burns.2020.04.039. Epub 2020 May 3. PMID: 33272743.
COPD exacerbation
[51] Scherr A, Graf R, Bain M, Christ-Crain M, Müller B, Tamm M, Stolz D. Pancreatic stone protein predicts positive sputum bacteriology in exacerbations of COPD. Chest. 2013 Feb 1;143(2):379-387. doi: 10.1378/chest.12-0730. PMID: 22922487.
Covid-19
[61] Van Singer M, Brahier T, Brochu Vez MJ, Gerhard Donnet H, Hugli O, Boillat-Blanco N. Pancreatic stone protein for early mortality prediction in COVID-19 patients. Crit Care 2021;25:1–4. doi:10.1186/s13054-021-03704-4.
Reviewer 2 Report
- Authors assessed novel protein biomarker, the pancreatic stone protein (PSP), as a biomarker of sepsis and questions its diagnostic and prognostic value in a review.
- Authors did not indicate the search engines and databases from where they obtained their data, the key words used, and when those searches were done.
- ethical approval and/or consent are not applicable
- methods and study design are not well described and not appropriate for answering the research question
- The methods, and materials, are not sufficiently detailed for reproduction research.
- statistical tests used are appropriate and correctly reported
- tables clear and accurately represent the results
- appropriate citations, support the claim made
- The results support the conclusions?
- limitations of the review not acknowledged
- the abstract is an accurate summary of the review and results,
- the language is clear and understandable
Author Response
1.“Authors did not indicate the search engines and databases from where they obtained their data, the key words used, and when those searches were done.”
We appreciate the reviewer’s comment. We have added a methods section to the manuscript and included a detailed description of the methods used to identify the references used.
2.“ethical approval and/or consent are not applicable”
We have removed the sentence regarding ethics approval from the manuscript and added the term “not applicable”
3.”methods and study design are not well described and not appropriate for answering the research question. The methods, and materials, are not sufficiently detailed for reproduction research.”
We recognize that the methodology was not properly detailed. We have now included a methods section and included the databases and search engines used, the key words and a PRISMA flow diagram to allow for reproduction research.
4.“The results support the conclusions?”
We believe that our conclusion are supported by the results, particularly those presented in tables 1 and 2. Since the publications exploring the applicability of PSP for the diagnosis and prognosis of infection/sepsis are scarce when compared to other established biomarkers (CPR and PCT), we believe we were cautious recognizing the limitations and highlighting the need for further studies.
5. limitations of the review not acknowledged
We have now added a section highlighting the limitations of the review.
Reviewer 3 Report
The paper is nice and well done. The tables are informative and clear. I suggest to includ a Figure on the study selection (see PRISMA reccomendations). Moreover, I suggest to include a short list with all clinical important key messages for the everyday practice.
Author Response
1” I suggest to includ a Figure on the study selection (see PRISMA reccomendations). “
We thank the reviewer for the suggestion. We have added a flow diagram on study selection as per PRISMA reccomendations
"2. Moreover, I suggest to include a short list with all clinical important key messages for the everyday practice. "
Once again we thank for the suggestion. We have included a shortlist of key messages at the end of the manuscript
Round 2
Reviewer 1 Report
Fine, let there be an updated review on PSP.